# Investigation on the Microscopic/Macroscopic Mechanical Properties of a Thermally Annealed Nafion^®^ Membrane

**DOI:** 10.3390/polym13224018

**Published:** 2021-11-20

**Authors:** Tuyet Anh Pham, Seunghoe Koo, Hyunseok Park, Quang Thien Luong, Oh Joong Kwon, Segeun Jang, Sang Moon Kim, Kyeongtae Kim

**Affiliations:** 1Department of Mechanical Engineering, Incheon National University, Incheon 22012, Korea; phamtuyetanh19@gmail.com (T.A.P.); shkoo7736@gmail.com (S.K.); qkrgustjr200@gmail.com (H.P.); 2Department of Energy and Chemical Engineering, Incheon National University, Incheon 22012, Korea; thienquang120396@gmail.com (Q.T.L.); ojkwon@inu.ac.kr (O.J.K.); 3School of Mechanical Engineering, Kookmin University, Seoul 02707, Korea; sjang@kookmin.ac.kr

**Keywords:** thermally annealing, Nafion^®^ membrane, atomic force microscopy, mechanical property, macro/microscopic modulus

## Abstract

The Nafion^®^ electrolyte membrane, which provides a proton pathway, is an essential element in fuel cell systems. Thermal treatment without additional additives is widely used to modify the mechanical properties of the membrane, to construct reliable and durable electrolyte membranes in the fuel cell. We measured the microscopic mechanical properties of thermally annealed membranes using atomic force microscopy with the two-point method. Furthermore, the macroscopic property was investigated through tensile tests. The microscopic modulus exceeded the macroscopic modulus over all annealing temperature ranges. Additionally, the measured microscopic modulus increased rapidly near 150 °C and was saturated over that temperature, whereas the macroscopic modulus continuously increased until 250 °C. This mismatched micro/macroscopic reinforcement trend indicates that the internal reinforcement of the clusters is induced first until 150 °C. In contrast, the reinforcement among the clusters, which requires more thermal energy, probably progresses even at a temperature of 250 °C. The results showed that the annealing process is effective for the surface smoothing and leveling of the Nafion^®^ membrane until 200 °C.

## 1. Introduction

As global environmental issues—including global warming and air pollution—get severe, renewable, and sustainable power sources have attracted significant attention. The most representative renewable energy sources are solar and wind; however, their electricity generation is often intermittent and fluctuates [1,2]. The extra electricity can be employed to generate hydrogen, and it can be used for providing sustainable electricity by using fuel cells. Along with this advantage, the polymer electrolyte membrane fuel cell is considered one of the most promising energy conversion devices due to its high-energy conversion efficiency with zero pollutant emissions, fast startup, and a low operating temperature [3,4,5]. Even with significant technological advances in fuel cell components, including the catalyst, electrolyte membrane, gas diffusion layer, and bipolar plates, it is still challenging to create high-performance, reliable, and durable fuel cells [6,7,8,9,10,11]. When it comes to the stability issue of the electrolyte membrane, though the widely used Nafion^®^ electrolyte membrane—which is constructed with a backbone of polytetrafluoroethylene (PTFE)—is considered to retain high chemical and mechanical stability, it is still inappropriate to use a Nafion^®^ membrane in the long term due to the instability of the perfluorinated sulfonic acid side chains, consecutive chemical attacks, and mechanical deformation during operation. Concerning the chemical stability, radicals such as H_2_O_2_, HO•, and HOO•, generated during the oxygen reduction reaction, deteriorate the membrane’s surface, leading to the formation of a pinhole [12,13,14,15]. This issue has been addressed and partially solved through a remarkable achievement: introducing radical scavengers, such as organic chemical species, inorganic nanoparticles, and metal ions, into the electrolyte membranes [16,17,18]. In addition, improving the mechanical properties of the membrane, which highly depend on the inherent material properties, has been extensively studied. Several studies exist on improving the mechanical properties by incorporating rigid materials, such as carbon nanotubes and graphene oxide [19,20,21,22]. The incorporated rigid materials can also act as a barrier to block certain undesired species from crossing over [23,24] and as mechanical reinforcements. However, suitable loadings of the additives and elaborate fabrication processes are required to avoid reduced proton conductivity and membrane cracking due to significant phase separation. As one of the methods for modifying mechanical properties without incorporating additional materials, thermal treatment of the pristine membrane is the most widely used process [25,26,27,28,29]. Until now, most studies have focused on investigating the bulk mechanical properties, such as tensile strength, Young’s modulus, and elongation to break. Meanwhile, microscopic studies of the mechanical property changes remain insufficient. Since operation fails due to the membrane’s degradation, which is attributed to the microscopic hotspot generation of the membrane, an investigation of the mechanical properties microscopically would be significant. Herein, the microscopic Young’s modulus and surface roughness of the thermally annealed membranes with annealing temperatures 100, 150, 200, and 250 °C were measured. The microscopic Young’s modulus was measured by atomic force microscopy (AFM) using the two-point method, and the surface roughness of an area of 300 × 300 nm was scanned to verify the surface-leveling effect of annealing. Furthermore, to confirm that the microscopic and macroscopic mechanical properties agreed, tensile tests were conducted to measure the macroscopic mechanical properties of the pristine and thermally annealed membranes. Additionally, this study deduced the morphological rearrangement of microscopic structures in the Nafion^®^ membrane based on a comparison of the changed microscopic and macroscopic mechanical properties of thermally annealed Nafion^®^ membranes.

## 2. Materials and Methods

### 2.1. Thermal Annealing Process for a Pristine Nafion^®^ Membrane

The annealing temperature was chosen considering the two glass transition temperatures (*T_g_*) of the Nafion^®^ membrane, which have been reported in previous studies [25,30]. The first *T_g_*, which ranges from 100 to 130 °C, induces the change in the mobility of the main chain, whereas the second *T_g_*, which is around 240 °C, changes the side chain of the polymer matrix [30,31,32]. Therefore, to analyze the effects of thermal annealing on the microscopic and macroscopic properties, the commercial Nafion^®^ membranes, N211, attached to the glass substrate, were annealed at temperatures of 100 °C (N211@100), 150 °C (N211@150), 200 °C (N211@200), and 250 °C (N211@250) for 2 h.

### 2.2. Measurements of Macroscopic Properties of Thermally Annealed Nafion^®^ Membranes through Tensile Test

To determine the mechanical bulk properties of the annealed and pristine membranes, tensile tests were performed at ambient temperature using a stretching machine (Instron, Norwood, MA, USA) with a controlled strain rate of 5 mm/min. The samples under examination were of the same size (10 mm width and 20 mm length). This test was also conducted to probe the relationships between the macroscopic and microscopic properties.

### 2.3. Measurements of Microscopic Mechanical Properties of Thermally Annealed Nafion^®^ Membranes through AFM

The microscopic properties of the pristine and annealed membranes at ambient conditions were measured using AFM (RHK, Troy, MI, USA). Additionally, the topography was measured using the AFM tapping mode with the same resolution of 256 × 256 pixels at a scan rate of 1 Hz. A deflection–displacement curve was obtained using a probe with a spring constant of 2.8 N/m, and the withdrawing speed of the probe from the substrate was set to be 100 nm/s. Additionally, Young’s modulus was obtained from the obtained deflection–displacement curve using the two-point method.

### 2.4. Membrane Electrode Assembly (MEA) Fabrication and Fuel Cell Testing

The MEA with the active area of 5 cm^2^ was fabricated by a spray coating method with the catalyst ink. The ink was prepared by mixing Pt/C 40 wt.% (Johnson Matthey, Devens, MA, USA) with deionized water, and Nafion^®^ solution 5 wt.% (DuPont, Wilmington, DE, USA) isopropyl alcohol (Sigma-Aldrich, St. Louis, MO, USA), and then further ultrasonicated for 20 min. The platinum loading was fixed at 0.2 mg/cm^2^ on both cathode and anode sides. After, the prepared MEA was placed between two gas diffusion layers (35BC, SGL Carbon, Wiesbaden, Germany), two Teflon gaskets, and two serpentine gas flow channels with a width of 1 mm. To evaluate the device’s performance, the single cell was connected to a PEMFC Test System (CNL Energy, Seoul, Korea) by supplying humidified H_2_ (300 mL/min) and air (1000 mL/min) to the anode and cathode, respectively, in the operating conditions of RH 92% and 70 °C. Then, high-frequency resistance (HFR) measurements were conducted at the same conditions as polarization tests with impedance spectroscopy (HCP 803, Biologic, Seyssinet-Pariset, France). HFR was measured at 0.5 V with an amplitude of 5 mV in the range of frequency from 1 kHz to 100 kHz.

## 3. Results and Discussion

As illustrated in Figure 1, pristine Nafion^®^ membranes were annealed at various temperatures—100, 150, 200, or 250 °C—2 h. The color of each the treated membranes was visually comparable to the pristine membrane (Appendix A). The slight color changes in N211@200 and N211@250 membranes were noted as extremely pale yellow compared to the inherent transparency of the pristine membrane. This phenomenon can be explained by the TGA measurement of Nafion^®^ membrane, as mentioned in previous studies [33,34,35,36], indicating that the dissociation of H_3_O^+^ occurred at 200 °C onwards.

### 3.1. Microscopic Properties of Pristine and Thermally Annealed Nafion^®^ Membranes

A two-point method with AFM was used to investigate the microscopic mechanical properties of the thermally annealed Nafion^®^ membranes. Since AFM measures the mechanical properties of the specific surface with a contact size of ~10 nm, which provides the highest spatial resolution, it is considered a precise approach for measuring morphology, adhesion energy, and Young’s modulus. With this exclusive advantage, AFM has been used to measure the alignment degrees of self-assembled monolayers [37], the adhesion energies of 2D materials [38], and Young’s moduli of polymers [39,40,41]. Therefore, AFM is the most suitable equipment for measuring microscopic mechanical features with the highest spatial resolution possible. Figure 2 shows a schematic illustration of measuring the microscopic Young’s moduli of the pristine and thermally annealed Nafion^®^ membranes (N211, N211@100, N211@150, N211@200, N211@250). The AFM measurements were conducted on randomly selected surfaces, each in a scan area of 300 nm × 300 nm. To calculate the microscopic Young’s modulus from a measurement, the two-point method was used. In the two-point method, the deflection of the cantilever and Piezo displacement in Z spectroscopy mode, which are converted to adhesion force and sample deformation, are measured at the two points. At a point, the adhesive force between the probe and substrate is zero (*F* = 0), whereas the other point exhibits the maximum adhesive force (*F* = *F_max_*) when the probe and substrate are separated (Figure 2b). Additionally, the microscopic Young’s modulus was estimated from the measured data at these two points. The deflection–displacement curve obtained by Z spectroscopy was converted into a force–deformation curve. With the converted values, the microscopic Young’s modulus was calculated based on JKR contact dynamics [42]. Following the JKR model, Young’s modulus (*E*) is expressed as follows:(1)$$E=\frac{3(1-\nu^{2})}{4}\times \frac{1.27F_{1}}{\sqrt{R{\left(\delta_{0}-\delta_{1}\right)}^{3}}}$$
where v is Poisson’s ratio, *R* is the radius of curvature of the tip, *δ* is the sample deformation, and *F*_1_ is the maximum adhesive force. A detailed process of calculating Young’s modulus from the deflection–distance curve is provided in the Appendix A). The radius of curvature of the tip was confirmed to be ~100 nm via a SEM observation (Figure 2c).

Before measuring the Young’s moduli of the Nafion^®^ membranes, the surface morphology of each sample was measured to confirm the surface smoothing and leveling effect of annealing. Figure 3 shows the AFM topography of the membranes measured in the tapping mode. In the 2D images in Figure 3, the surface roughness decreases as the annealing temperature increases. To quantitatively compare the surface roughness for each sample, the cross-sectional topography was plotted following the arbitrary line on the 2D image, and the measured mean surface roughness was 0.857 nm for N211, 0.524 nm for N211@100, 0.444 nm for N211@150, 0.244 nm for N211@200, and 0.355 nm for N211@250. Thus, it was confirmed that the annealing process was effective for surface smoothing and leveling of the Nafion^®^ membrane until 200 °C. However, above 200 °C, the leveling effect was analogous. Compared with N211, the surface roughness of N211@200 decreased by 28%. Additionally, the ~30 nm-sized grains on the pristine N211 gradually decreased in number as the annealing temperature increased, implying the reconstruction of grains due to annealing.

Figure 4 and Table 1 show the microscopic Young’s moduli of pristine and thermally annealed Nafion^®^ 211 membranes, which were obtained using AFM with the two-point method. The measured Young’s modulus of the pristine Nafion^®^ membrane was ~372 MPa. As the annealing temperature increased, the microscopic Young’s modulus of the treated membrane increased compared to that of the pristine membrane. The ascending trend of Young’s modulus corresponds to the descending trend of the surface roughness. The Young’s modulus rapidly increased at annealing temperatures between 100 and 150 °C, which corresponds to the first *Tg* of ~120 °C. Besides, the Young’s modulus seems to have been saturated from the annealing temperature of 150 °C, with a value of ~745 MPa. Even though the annealing temperature exceeded the second *Tg* of ~240 °C, the Young’s modulus of N211@250 was analogous to that of N211@150, implying that the mobility change of the side chains does not severely affect this microscopic mechanical property. It can be inferred that the Young’s modulus is reinforced due to the rearrangement of the grains of the Nafion^®^ membrane, which is strongly related to the change in the main chain of the Nafion^®^ membrane.

### 3.2. Macroscopic Mechanical Properties of Pristine and Thermally Annealed Nafion^®^ Membranes

A uniaxial tensile test was conducted for each membrane to confirm whether these reinforcements applied to the macroscopic properties. The stress–strain behavior of the membrane, including the Young’s modulus from the initial slope in the elastic region, tensile strength, and elongation to break, were investigated by calculating the strain and stress considering the sample length and area [43]. Figure 5 shows the enhancement in the stress–strain responses of the pristine and annealed membranes as the annealing temperature increases. The Young’s modulus of the pristine membrane was 208.69 MPa; after annealing at 100 °C, it was 210.02 MPa. The Young’s modulus was increased by 5.42%, 8.08%, and 14.07% for N211@150, N211@200, and N211@250 compared to that of N211, respectively. A similar trend was found in the measured tensile strength. N211 displayed the lowest value of 17.14 MPa, which was 22.69% and 27.84% lower than those of N211@100 and N211@150, respectively. As illustrated in Figure 5c, the tensile strengths of N211@200 (34.23 MPa) and N211@250 (34.69 MPa) were approximately double that of the pristine membrane. Annealing also affected the elongation to break of the membrane. There were similar values for N211@100, N211@150, and N211@200 membranes, which were ~1.8-fold higher than that of the pristine membrane (175%). N211@250 slightly decreased in elongation to break with a value of 243% compared to N211@200. Accordingly, annealing is important in improving the mechanical properties of membranes by promoting the strengthened interactions among Nafion^®^ polymer chains, and a higher degree of crystallinity in the polymer matrix [35,44].

### 3.3. Relationship between Microscopic and Macroscopic Modulus

To compare the ascending trend of Young’s moduli at the microscopic and macroscopic levels for the pristine and thermally annealed membranes, the Young’s moduli for each case are plotted on a single graph (Figure 6). The graph shows that the microscopic modulus exceeded the macroscopic modulus in all cases. For the pristine membrane, the microscopic modulus was measured as 372 MPa, which is 78.25% higher than the macroscopic modulus. When the annealing temperature increased to 150 °C, the modulus difference was the largest found—3.3-fold. In contrast, the difference in moduli decreased as the annealing temperature exceeded 200 °C. Interestingly, the microscopic modulus increased rapidly near 150 °C and saturated above it afterward. However, the macroscopic modulus continuously increased from 100 to 250 °C. The increases in macroscopic modulus appeared prominently near 150 and 250 °C, corresponding to the first and second *Tg* of the Nafion^®^ membrane. Based on the microscopic and macroscopic modulus changes, it was inferred that the Nafion^®^ membrane comprises spatially separated microscopic Nafion^®^ clusters, and the collection of the microscopic clusters forms the bulk of the membrane (wherein there would be boundaries between clusters. If so, the mechanical properties can be explained in two ways: internal intermolecular adhesion forces within the microscopic clusters and interfacial adhesion forces at the boundaries of the clusters. Since the AFM measurement covers the nanosized area of a membrane, it can be considered that the microscopic AFM measurement reflects the internal adhesion force within the microscopic clusters, and the bulk tensile test reflects the interfacial adhesion force at the boundaries of the clusters. Then, it can be inferred that the annealing near 150 °C induced the reinforcement of the internal clusters via the rearrangement of the main chain orientation in the polymer matrix [45,46] and crystallinity of the PTFE backbone [26,35,47], and this change seemed mostly progressive below 200 °C. This trend is similar to the increased crystallinity as the annealing temperature increased, which was confirmed through X-ray diffraction (XRD) measurements in previous studies [28,35]. Concerning the macroscopic moduli of the membranes, the reinforcement between the clusters was caused by increased crystallization between the clusters until 250 °C. Since it would require more thermal energy to rearrange the relative bulk clusters than the microscopic molecules in the cluster, a higher annealing temperature is necessary to improve the macroscopic modulus. The results can be explained by the surface morphology shown in the Figure 3. When comparing the local grain size and roughness, which are indicated as red dotted lines in the topography and cross-sectional graph, the local grains size of the pristine Nafion membrane was found to be ~30 nm. Additionally, while increasing annealing temperature up to 150 °C, the clusters in the grains combined, and the local grains were finely split. At the annealing temperatures above 150 °C, there is no significant change in grain size (~5 nm), which matches with the ascending trend of the measured microscopic Young’s modulus. In the aspect of structural groove, the vertical gap of the grooves was continuously reduced as the annealing temperature increased until 250 °C, and the boundary between the grains gradually became indistinct. This result matches well with the ascending trend of the measured macroscopic Young’s modulus.

Finally, to confirm the feasibility of the thermally annealed membranes for fuel cell operation, membrane electrode assemblies for each membrane were constructed and electrochemically characterized. Appendix A shows that MEA with N211@200 exhibited comparable performance to that with a pristine N211 membrane, which indicates that there were no serious chemical properties changes in the treated membrane, whereas performance of MEA with N211@250 decreased significantly. The results indicate that the annealing temperature over 250 °C induced the destruction of the membrane, leading to PEMFC operation failure. In addition, the ionic conductivity of the membranes was characterized through measurement of high-frequency resistance (HFR), which represents the membrane ionic resistance, since ionic conductivity is inversely proportional to the ohmic loss of the membrane (Appendix A). The results indicate that although a slight change (~3.04 %) in ionic conductivity compared to the pristine Nafion membrane was induced due to the thermal annealing at 200 °C, the modified membranes could be properly applied to fuel cells, except for N211@250. The results imply that annealing temperatures below 200 °C are feasible for membranes used for fuel cells, but an annealing temperature over 250 °C induces destruction of the membrane, which can lead to fuel cell failure. 

## 4. Conclusions

Herein, the microscopic and macroscopic mechanical properties of pristine and thermally annealed Nafion^®^ membranes were investigated. The annealing temperatures of the membranes were 100, 150, 200, and 250 °C. The microscopic and macroscopic mechanical properties were characterized through AFM and tensile tests, respectively. It was confirmed that the surface roughness of the thermally annealed membranes with an annealing temperature of 200 °C decreased by 28%. The microscopic modulus rapidly increased to ~734 MPa near the annealing temperature of 150 °C, and was saturated over that temperature, whereas the macroscopic modulus continuously increased over the whole the range of annealing temperature. Precisely, the increments in modulus were apparent near 150 °C and 250 °C, which are near the first and second *T_g_* of the Nafion^®^ membrane. Furthermore, the difference in microscopic and macroscopic modulus was the largest near 150 °C and reduced past that temperature. Based on the assumption that the microscopic and macroscopic moduli of the membranes result from internal adhesion forces in the clusters and the interfacial adhesion forces between the clusters, respectively, the results show that the reinforcement of the molecules in the cluster occurs first within the annealing temperature of 150 °C due to the rearrangement of the main chains. Additionally, the cluster rearrangement, which requires higher thermal energy, progresses until 250 °C, which enhances the bulk modulus. Finally, based on the fuel cell testing with each membrane, applying an annealing temperature below 200 °C to the membrane is appropriate with fuel cells use in mind, and an annealing temperature over 250 °C induces destruction of the membrane, which can lead to fuel cell failure.

## Figures and Tables

**Figure 1 polymers-13-04018-f001:**
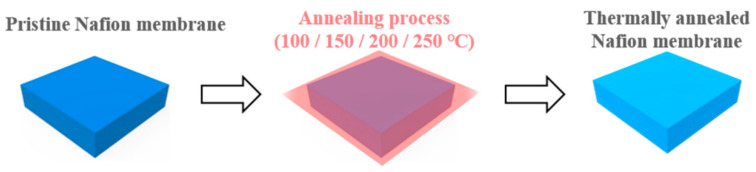
Schematic illustration of annealing treatment on pristine Nafion^®^ membrane at various temperatures, 100 °C (N211@100), 150 °C (N211@150), 200 °C (N211@200), 250 °C (N211@250), for 2 h.

**Figure 2 polymers-13-04018-f002:**
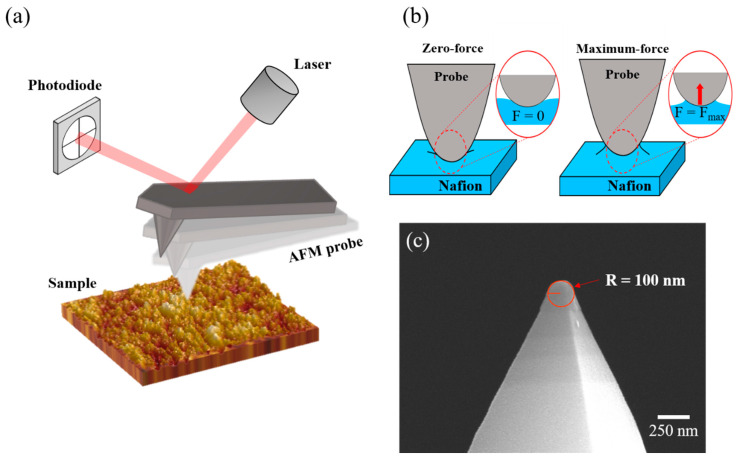
(**a**) A schematic of the measurement of Young’s modulus of pristine and thermally annealed Nafion^®^ membranes through Z spectroscopy mode of an AFM. (**b**) A schematic of the two-point method to obtain Young’s modulus, which is calculated by measuring deformations at zero-force (*F* = 0) and maximum-force (*F = F_max_*) points [42]. (**c**) SEM image of the end of the AFM probe tip used.

**Figure 3 polymers-13-04018-f003:**
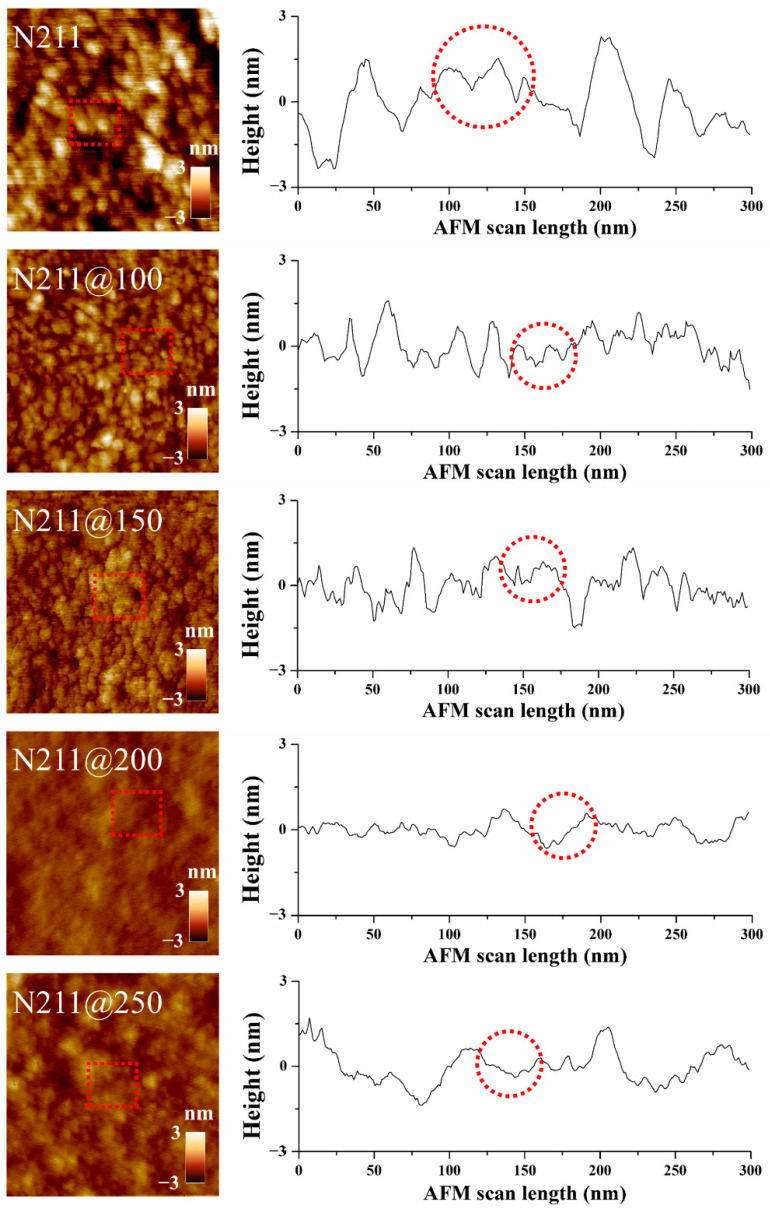
AFM topography measured in tapping mode of pristine and annealed membranes (**left**), and corresponding cross-sectional topography (**right**).

**Figure 4 polymers-13-04018-f004:**
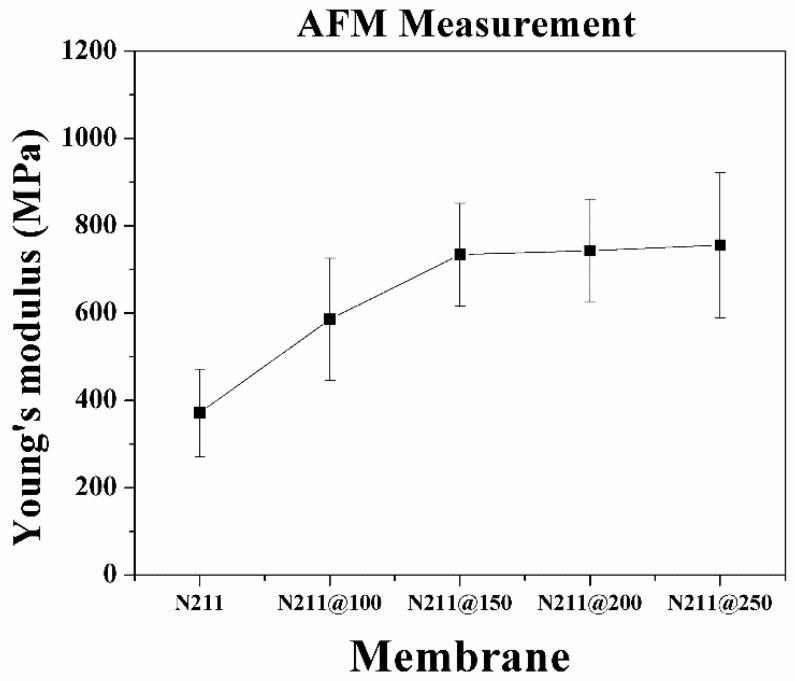
Measured microscopic Young’s moduli of pristine and thermally annealed Nafion^®^ 211 membranes using AFM.

**Figure 5 polymers-13-04018-f005:**
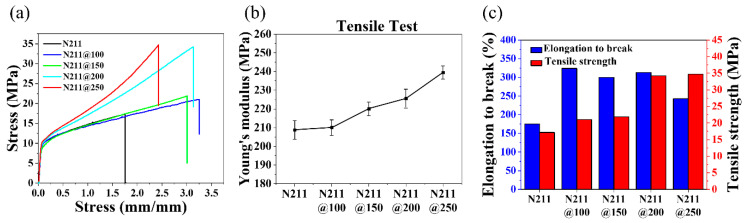
(**a**) Stress–strain curves of pristine and annealed membranes. (**b**) Plot for measured Young’s moduli of pristine and annealed membranes. (**c**) Plot for elongation to break values and tensile strengths of pristine and annealed membranes.

**Figure 6 polymers-13-04018-f006:**
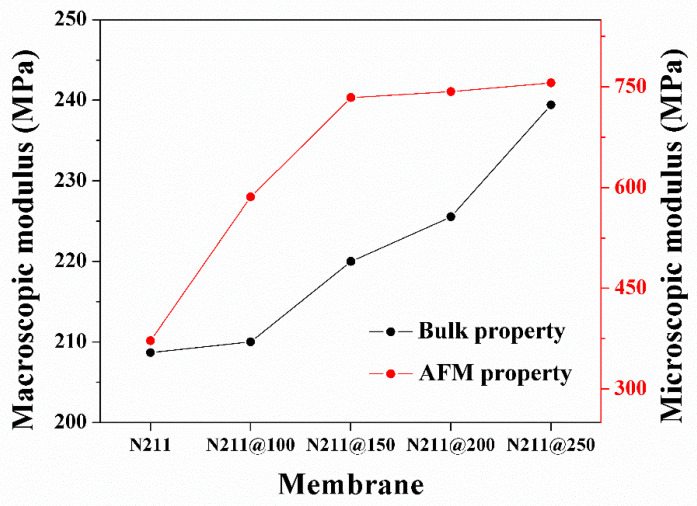
Microscopic and macroscopic Young’s moduli for pristine and thermally annealed membranes.

**Table 1 polymers-13-04018-t001:** Average value and standard deviation of Young’s modulus measurement data for each membrane.

Annealing Temperature	Young’s Modulus (MPa)	Standard Deviation (%)
No annealed	372	26.94
100 °C	586	23.95
150 °C	734	16.04
200 °C	743	15.83
250 °C	756	22.14

## Data Availability

Not applicable.

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
