# Peer review of "Investigation on the Microscopic/Macroscopic Mechanical Properties of a Thermally Annealed Nafion® Membrane"

_polymers, 2021, doi:10.3390/polym13224018_

Round 1
Reviewer 1 Report
Tuyet Anh Phama et al reported, “Investigation on microscopic/macroscopic mechanical proper- 2 ties of thermally annealed Nafion® membrane”. In this report author mentioned the intersecting results of the mechanical properties of pristine and thermally annealed Nafion®membranes samples.
All reported results are good and positive for readers. After reading of whole paper, I strongly recommend with minor suggestion.
- Author should corrected as standard format of temperature 150°C, as 150 °C.
- Herein, the microscopic Young’s modulus and 61 surface roughness of the thermally annealed membranes with annealing temperature variation of 100°C, 150°C, 200°C, and 250°C were measured, Author should corrected standard format of unit of the temperature, as 100, 150, 200, and 250 °C.
- Conclusion section is too long, Author should be revised.
- Quality of the all figure should be revised.
- Correct all figure in the same format in whole paper.
- Author should use surface morphology of the detail study of the pristine and thermally annealed Nafion® membranes samples.
Author Response
We respond to the attached file. please find the response file.

Reviewer 2 Report
In this manuscript, the microscopic mechanical properties of thermally annealed Nafion membranes using atomic force microscopy, while the macroscopic property was investigated through tensile tests. It is found the microscopic modulus exceeds the macroscopic modulus over all temperature ranges. The mismatch between micro/macroscopic reinforcement trends was investigated. The results show that the annealing process is effective for surface smoothing and levelling of the Nafion membrane.
I consider the content of this manuscript will definitely meet the reading interests of the readers of the Polymers journal. However, the discussion is still slightly monotonous and confusing, and the introduction needs to be further improved. The unique meaning of this work needs to be emphasized, by double-checking the membrane ion conductivity after annealing, or supplying the fuel cell polarization curves for the membranes after annealing.
Therefore, I suggest giving a minor revision and the authors need to clarify some issues or supply some more data to enrich the content.
- Abstract and Introduction
For the Keywords, ‘macroscopic’, ‘microscopic’, and ‘modulus’ can be further added to attract the reading interests of the readers.
Line 30, the authors should better explain why sustainable power sources have attracted significant attention, which indicates the disadvantages of renewable sources.
Because ‘most of the renewable energy sources are intermittent, opening spatial and temporal gaps between the availability of the energy and its consumption by end-users. In order to address these issues, it is necessary to develop suitable energy storage systems for the power grid [Electrochimica Acta 309 (2019): 311-325; Reactive and Functional Polymers 157 (2020): 104777]’.
Line 38, ‘Especially, electrolyte membrane stability, which acts as a proton pathway and insulation of electron transfer between the anode and cathode, is one of the most critical issues.’
This sentence is not carefully written and is confusing. The stability of the membrane is important for the long-term operation of fuel cells for sure. But it is not ‘membrane stability’ that acts as the proton pathway and insulation of electrons transfer, but the ‘electrolyte membrane’ itself.
Line 44, considering the chemical stability of Nafion, the description is not that accurate:
- About the structure, the PTFE backbone is very chemically stable and known as Teflon, and the chemical stability issue does not only come from the backbone. Therefore, the structure should be further explained, especially about the perfluoroethereal side chains and the hydrophilic sulfonic acid group, which is less chemically stable compared to the PTFE backbone (10.1016/j.ssi.2018.01.038).
- The description‘radicals, such as H2O2, HO•, and HOO•, generated during the oxygen reduction reaction, deteriorate the membrane surface, leading to the formation of the pinhole’ is doubtful. It is well-known that Nafion is generally pretreated with 3 wt% H2O2 solutions and 1M H2SO4 at elevated temperatures, but why there is no formation of the pinhole during the pretreatment?
Line 47, ‘organic chemical species, inorganic nanoparticles, and metal ions’ can be used to solve the chemical stability problem. While Line 52, ‘carbon nanotubes and graphene oxide’ improves the mechanical stability but ‘block the proton pathway and reduce conductivity’.
These descriptions are partially correct, but it is not comprehensive description. Indeed, no matter inorganic particles or carbon materials, too high loading of incorporation into the polymer membrane will block the proton pathway (even make the membrane crack due to significant phase separation). But with suitable (lower than 20 wt%) loadings, they can improve the species selectivity of the membranes by inhibiting the crossover of undesired species, such as methanol in DMFCs [Journal of Membrane Science 541 (2017): 127-136] and metallic species for flow batteries [Electrochimica Acta 378 (2021): 138133].
Generally, the species selectivity is improved by sacrificing proton conductivity slightly to prevent undesired crossover as much as possible (composite membrane acts as a barrier layer to block certain undesired species). So it is not to say that the addition of carbon or inorganic materials only brings about blocking problems without any other benefits, and the importance is to select suitable loadings of additives.
- Materials and Methods
Line 74, ‘The annealing temperature was chosen considering the two glass transition temperatures’.
It is understandable that based on two Tg, the annealing temperature is determined. But attention should be paid that these factors are more related to the physical properties of the Nafion polymer. However, the chemical properties should also be considered as well. It is widely reported by TGA analysis of Nafion membranes that ‘between ca. 130 and 250 °C starts with the loss of the –SO3H groups’. So I doubt whether the annealing temperature of 200 °C and 250 °C for 2 h is too high for the polymers, and indeed the membrane has degraded. Did the author check the ionic conductivity of the treated membrane material again? It may not be the focus of this paper, but if the chemical properties have changed, the membrane may no longer be suitable for practical applications anymore.
- Results and Discussion
Line 108, ‘indicating that the dissociation of H3O+ occurred from 200°C onwards, which could merely affect the change of the membrane. ’
It is not correct. ‘in the 170-250 °C temperature region, corresponds to the decomposition of -SO3H groups’. [see Figure 1 of The Journal of Physical Chemistry B 110.49 (2006): 24972-24986]. If -SO3H groups are stable until 200 °C, why do high-temperature PEMFCs adopt PBI membrane, not Nafion membrane directly? I reiterate that the ionic conductivity of the membrane after annealing should be retested to avoid confusion and misunderstanding.
Line 238, ‘ Since it would require more thermal energy to rearrange the relative bulk clusters than it does to the microscopic molecules in the cluster, a higher annealing temperature is necessary to improve the macroscopic modulus.’
I disagree with this description. If an even higher temperature is used for annealing at 300 °C, I consider both the -SO3H and side chains will be destroyed.
In addition, since this treatment is for the applications of the fuel cells, why membranes are not tested with fuel cell in-situ polarization curves? Even without a conductivity test before and after annealing, if the membranes after annealing exhibit high performance during the fuel cell test, it also indicates the treatment temperature is reasonable. But now only mechanical properties and morphology are shown in the paper, without further verification of ion-conducting related properties/performance can be found.
Author Response

(The authors gave the same response as above.)
